# Child marriage in Canada: A systematic review

**Michele Zaman****[1], Alissa Koski[2]** *

1 Master of Science in Public Health student, McGill University, Department of Epidemiology, Biostatistics and Occupational Health, Montreal, Quebec, Canada, 2 Assistant Professor, McGill University Department of Epidemiology, Biostatistics and Occupational Health and the Institute for Health and Social Policy, Montreal, Quebec, Canada

* alissa.koski@mcgill.ca

## Abstract

Child marriage, defined by the United Nations as marriage before the age of 18, is considered a violation of human rights with negative consequences for girls' health. We systematically reviewed existing academic literature and news media to learn what is known about the frequency of child marriage in Canada and its effects on health. Approximately 1% of 15-19-year-olds in Canada were married or in common law unions in 2016. News reports document cases of child marriage among religious minority communities but no nationwide estimates of the frequency of marriage before the age of 18 were identified. Sources consistently show girls are more likely to marry as teens than boys. Information on married teens between 15 and 19 years of age suggests similarities in marriage patterns among this age group in Canada and child marriage practices globally. Further research is needed to measure Canada's progress toward eliminating child marriage.

**Data Availability Statement:** The data used for our results tables are all from our included articles and or publicly available data from Statistics Canada. Statistics Canada, 2016 Census of Population, Statistics Canada Catalogue no. 98-400-X2016028.

## Introduction

The Office of the United Nations High Commissioner for Human Rights defines child marriage as the formal or informal marriage of any person under the age of 18 [1]. The practice is widely considered a violation of human rights that harms the health and development of children, most often girls [1]. Over the past sixty years, numerous human rights instruments have addressed child marriage. The 1962 United Nations Convention on Consent to Marriage, Minimum Age for Marriage and Registration of Marriages required member countries to establish a minimum age for marriage, though no explicit statement on what the minimum should be was made at that time [2]. The Convention on the Elimination of All Forms of Discrimination against Women states that the "marriage of a child shall have no legal effect" and reiterates the need for member countries to establish a minimum age for marriage, but again did not recommend a specific age [3]. A decade later, the Convention on the Rights of the Child defined a child as any person under the age of 18 and since then the proportion of women marrying before that age has been used as a quantifiable indicator of international development [4]. The United Nations Sustainable Development Goals, adopted in 2015, call for the elimination of child marriage by the year 2030 and estimates of the proportion of women who married before the age of 18 are included in annual reports of progress toward that goal [5].

Additionally, our search strategy for the systematic review is included as an appendix.

**Funding:** Funding for this research was received by AK, she received the Social Sciences and Humanities Research Council (SSHRC), Insight Development Grant no. 430-2018-00481. More information on the insight development grant can be found here: https://www.sshrc-crsh.gc.ca/funding-financement/programs-programmes/insight_development_grants-subventions_de_developpement_savoir-eng.aspx. The funders did not play any role in the study design, data collection and analysis, decision to publish or preparation of the manuscript.

**Competing interests:** The authors have declared that no competing interests exist.

Despite these efforts, child marriage continues across the globe and is associated with a range of negative health outcomes that have repercussions throughout the life course. Girls who marry before the age of 18 begin having children of their own at early ages. Research from Africa and South Asia shows that they have less control over their own fertility: they report having less access to contraception and more unwanted pregnancies, and give birth at shorter intervals when compared to girls who marry at later ages [6, 7, 8]. This may reflect limited agency to negotiate the frequency of sex or the use of contraception within these unions, particularly when there is a large age discrepancy and/or power imbalance between spouses [8]. These factors put them at elevated risk of obstetric complications, the leading cause of death among adolescent girls in low and middle-income countries [9]. Girls who marry before 18 years of age are also more likely to contract HIV and to experience domestic violence and mental health disorders [10, 11, 12]. Though the vast majority of attention to child marriage and its harms is focused on Africa and South Asia, the practice continues in wealthy countries as well [13]. Research on the consequences of early marriage in these contexts is scarce, but available evidence suggests that it remains harmful to health despite stronger public health infrastructure and health care systems. The children of adolescent mothers in the United States are more likely to be born pre-term, at low birth weights, and to die during the first 28 days after birth, even after controlling for socioeconomic factors that often confound this relationship [14]. American women who married before the age of 18 were also more likely to report experiencing substance abuse and mental health disorders as adults [15]. It is important to note that establishing whether child marriage actually causes these adverse events is challenging. It is possible that girls who marry as children differ from their peers who marry later with regard to other characteristics that may put them at risk of these outcomes. For example, growing up in poverty may place children at greater risk of marriage and at greater risk of mental health disorders in adolescence, independent of their marriage.

In addition to adverse effects on health, child marriage is also strongly associated with educational and economic outcomes. Across the globe, girls who marry before the age of 18 have less schooling than their peers who marry later [15, 16, 17, 18] but the directionality of this relationship probably varies across contexts. Research from Bangladesh and Uganda shows that child marriage leads to early departure from school [16, 18] However, in contexts where girls' access to schooling is restricted as a result of poverty, physical proximity, or other factors, child marriage is unlikely to be a cause of low educational attainment and may instead be perceived as an alternative means of transitioning to adulthood [19, 20]. Research from the United States shows that girls who married before the age of 16 were 31 percentage points more likely to live in poverty in adulthood when compared to girls who married at later ages [17].

Child marriage remains legal in many countries despite commitments to end the practice and consistent evidence that it is detrimental for health and well-being [21]. The Canadian *Civil Marriage Act* was updated in 2015 to prohibit marriage before the age of 16 years, but 16 and 17-year-olds can legally marry with parental consent [22]. Canadian law on this issue is at odds with the country's efforts to end child marriage elsewhere in the world through foreign policy. Canada has co-sponsored United Nations resolutions to end child marriage and provides financial resources for this purpose through its Feminist International Assistance Policy [23, 24]. Given the country's efforts to address child marriage on a global stage, we believe it is important to apply the same definitions and metrics used in low and middle-income countries to domestic practices. We undertook this systematic review to learn what is already known about child marriages in the country, including motivations for the practice and its context, its consequences, and how frequently it occurs.

## Methods

We conducted a systematic review of academic literature and media sources to identify any published information on the prevalence, motivations for, or consequences of early marriages that occurred within Canada. We searched three distinct databases (Scopus, ProQuest including Canadian Newsstream, and Google Scholar) for articles that referred to early, adolescent, teen, child or underage marriage, with key terms for Canadian provinces and territories. The detailed search terms were developed in consultation with a research librarian at McGill University and are included in the appendix. We conducted our searches in March 2019 and results were not restricted in terms of publication dates.

Searches of all three databases returned a total of 941 citations: 249 from Scopus, 514 from ProQuest and Canadian Newsstream, and 178 from Google Scholar (Fig 1). A total of 128 duplicate citations were identified, leaving a total of 813 unique citations. The title and abstract (if available) of each unique citation was screened by a single reviewer (MZ) according to pre-defined inclusion and exclusion criteria. Citations were brought forward for full-text review if the title and abstract were written in English or French, if the article was published in an academic or professional society journal or a newspaper, and if they included qualitative or quantitative information on early marriages that occurred in Canada or case reports of early marriages. Citations were excluded at this stage if they referred to conference abstracts or blog posts, if they referred solely to early marriages that occurred outside of Canada or considered only marriages to persons aged 25 years and older. We extended the age cutoff for this review beyond 18 years in an effort to include any studies of early marriage in Canada, even if an age threshold other 18 years was used. If the precise age range of the marriages under consideration was not clear based on review of the title and abstract, for example, if the article referred to adolescent marriages without specifying an age range, the article was brought forward for full-text review.

Initial screening of titles and abstracts yielded 108 items for full-text review. Two reviewers (MZ and AK) read the full text of each of these articles and made independent decisions regarding their inclusion or exclusion based on the same criteria described above. We identified an additional 8 citations of potential relevance by manually searching the bibliographies of articles included in the full-text review and assessed these for eligibility as well. Discrepancies between the two reviewers were resolved through discussion. We were unable to find the full text of 3 of the news sources identified in our search despite the assistance of a research librarian at McGill University.

We identified a total of 27 articles that met our inclusion criteria. Fig 1 illustrates the process of our systematic review. Nearly all of these articles (n = 24) were newspaper reports of individual marriages or descriptive analyses of trends in the frequency of marriage among different age groups based on data collected by Statistics Canada. Only 2 studies were designed to estimate the effect of early marriage on specified outcomes and 1 estimated the effect of minimum wage increases on teen marriage rates. We discuss the limitations of each of these 3 studies in our results but we did not systematically quantify the extent of bias because the number of etiologic studies identified was so small. The presence or magnitude of publication bias could not be reliably assessed for the same reason.

## Results

Sixteen of the 27 articles included in our review were published in newspapers across Canada between 1983 and 2017 (Table 1).

Nearly all of them are case reports of marriages that occurred among small religious groups in British Columbia and Quebec. Eight of the articles refer to the marriage of girls under the

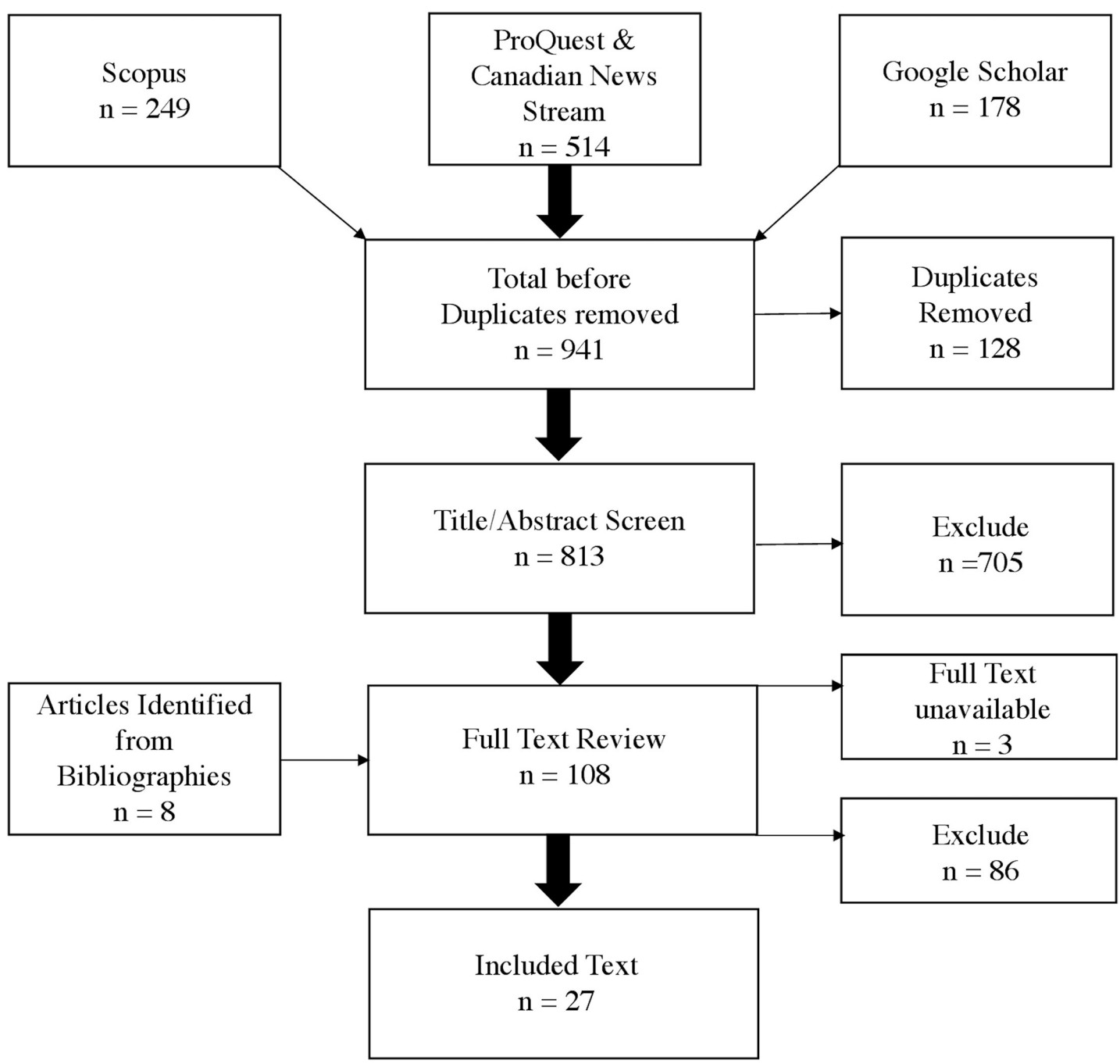

**Fig 1. Diagram detailing the flow of information through phases of the systematic review.**

age of 18 among the Fundamentalist Church of Latter Day Saints community based in Bountiful, British Columbia. News reports on legal proceedings involving the community's religious leaders drew attention to cases in which 15 and 16-year-old girls were wed. These reports included statements from women who testified that they were married before the age of 18 and describe court documents that attest to the marriages [25, 28, 30]. Five of the newspaper articles refer to a single legal case in Quebec in which the 52-year-old leader of a Christian

**Table 1. Details on the 27 articles on early marriage in Canada identified through systematic review of academic literature and media sources.**

| Author(s) | Title | Year Published | Source | Information on early marriage in Canada |
|---|---|---|---|---|
| *Newspaper Sources* | | | | |
| Bramham, D [25] | Exploitation is out in the Open | 2017 | Vancouver Sun | Acknowledgement of girls married before the age of 18 among the Fundamentalist Church of Latter Day Saints community in Bountiful, British Columbia |
| Bramham, D [26] | Trial might only be the first for polygamous community | 2016 | Vancouver Sun | Acknowledgement of girls married before the age of 18 among the Fundamentalist Church of Latter Day Saints community in Bountiful, British Columbia |
| Woods, A [27] | Jewish Sect Says Raids Targeted Child Marriages | 2014 | The Toronto Star | Statements regarding the marriage of children under the age of 18 among a Jewish sect in Quebec. |
| Bramham, D [28] | Terrible Choice: child bride or plural wife? | 2011 | Daily Bulletin, British Columbia | Acknowledgement of girls married before the age of 18 among the Fundamentalist Church of Latter Day Saints community in Bountiful, British Columbia |
| Not listed [29] | RCMP Investigates Trafficking of Girls from Mormon Community to the U.S. | 2011 | The New Glasgow News | Allegations that girls under the age of 18 were brought from the United States to Canada to wed among the Fundamentalist Church of Latter Day Saints in Bountiful, British Columbia |
| Kellar, J [30] | Court Documents Link Polygamist Blackmore to Cross-Border Marriages | 2011 | The Canadian Press | Acknowledgement of girls married before the age of 18 among the Fundamentalist Church of Latter Day Saints community in Bountiful, British Columbia |
| Matas, R [31] | Crackdown at the Canadian Border urged | 2008 | The Globe and Mail | Allegations that girls under the age of 18 were brought from the United States to Canada to wed among the Fundamentalist Church of Latter Day Saints in Bountiful, British Columbia |
| Bramham, D [32] | Who are these "sister-wives" Protecting? | 2005 | The Vancouver Sun | Acknowledgement of girls married before the age of 18 among the Fundamentalist Church of Latter Day Saints community in Bountiful, British Columbia |
| Sanders, C [33] | Unlikely Bedfellows Advocate Raising Age of Sexual Consent | 2005 | CanWest News | Acknowledgement of girls married before the age of 18 among the Fundamentalist Church of Latter Day Saints community in Bountiful, British Columbia |
| Hanes, A [34] | Preacher Who Married 10-year-old says their love story has been twisted | 2005 | Calgary Herald | Coverage of a legal case in Quebec involving the alleged marriage of a 52-year old religious leader to a 10-year old girl. |
| Not listed [35] | Child Bride Had No Say, Social Worker Testifies | 2004 | National Post | Coverage of a legal case in Quebec involving the alleged marriage of a 52-year old religious leader to a 10-year old girl. |
| Block, I [36] | 'Bride' Wed to Pastor at Age 10 | 2004 | Sault Star | Coverage of a legal case in Quebec involving the alleged marriage of a 52-year old religious leader to a 10-year old girl. |
| Hanes, A [37] | Man, 52 Accused of Abuse for Marriage to 10-year-old | 2004 | Kingston Whig-Standard | Coverage of a legal case in Quebec involving the alleged marriage of a 52-year old religious leader to a 10-year old girl. |
| Hanes, A [38] | 'Hubby' Accused of Abuse | 2004 | Leader Post | Coverage of a legal case in Quebec involving the alleged marriage of a 52-year old religious leader to a 10-year old girl. |
| Langton, M [39] | She's Designing her Future | 1997 | The Spectator | A personal interest story about a woman who became pregnant and married at 18 and now is running her own clothing line. |
| Not listed [40] | Child Marriages Up Last year, But Still Rare | 1983 | Calgary Herald | Coverage of a government report indicating that 29 children under the age of 15 were married in Canada in 1982. |
| *Academic sources and grey literature* | | | | |
| Johnson, MD et al. [41] | Better Late Than Early: Marital Timing and Subjective Well-Being in Midlife | 2017 | Journal of Family Psychology | The authors report that persons from a cohort of high-school graduates in Edmonton, Alberta who married before the age of 23 reported more depressive symptoms in their early 40s when compared to those who married at later ages. |

*(Continued)*

**Table 1.** (Continued)

| Author(s) | Title | Year Published | Source | Information on early marriage in Canada |
|---|---|---|---|---|
| Morin, KB [42] | Understanding the Practice of Early and Forced Marriage in North America and its Effects on Women | 2017 | Samuel Centre for Social Connectedness | The report includes results of qualitative interviews with women who married early, including one from Quebec. |
| Statistics Canada [43] | Family Characteristics of Adults, Age and Sex for the Population 15 years and Over in Private Households | 2017 | Statistics Canada catalogue number 98-400-X2016028 | The official results from the 2016 Canadian census, including the number of 15-19-year-olds married or in common law unions by gender and province/territory |
| Sen, A and Ariizumi, H [44] | Teen families, welfare transfers, and the minimum wage: evidence from Canada | 2013 | Canadian Journal of Economics | The authors examined the association between changes in minimum wage and marriage and birth rates among 15-19-year-olds across Canada. |
| Gompf, K [45] | Child brides and shining a light | 2005 | Relational Child and Youth Care Practice | The author describes his conversation with a young married girl from Bountiful, British Columbia. |
| Romaniuc, A and Chuiko, L [46] | Matrimonial Behaviour in Canada and Ukraine: The Enduring Hold of Culture | 1999 | Journal of Comparative Family Studies | The authors present historical trends in early marriage in Canada between 1891 and 1991. |
| Gentleman, J and Park, E [47] | Age Differences of Married and Divorcing Couples | 1994 | Health Reports | The authors examine age differences between spouses using data from the 1991 Canadian census. |
| Wadhera, S and Strachan, J [48] | Demographic Trends in Marriages in Canada: 1921–1990 | 1992 | Health Reports | The authors present marriage rates among 15-19-year-olds over time. |
| Wadhera, S and Strachan, J [49] | Marriages in Canada, 1991 | 1992 | Health Reports | The authors present marriage rates among 15-19-year-olds. |
| Lapierre, L [50] | Divorces, Canada and the Provinces, 1990 | 1991 | Health Reports | The author presents divorce rates among 15-19-year-olds. |
| Grindstaff, CF [51] | Adolescent Marriage and Childbearing: The Long-Term Economic Outcome, Canada in the 1980s | 1988 | Adolescence | The author examines the educational attainment and other economic outcomes among women who married before the age of 20. |

religious group was alleged to have married a 10-year-old girl. The published reports indicate that the girl testified that she was molested by the man who claimed to be her husband [37, 38]. Another article included a statement from a former member of a small Jewish group who claimed that he was married to a 15-year-old girl when he was 25 years of age and that the ceremony was arranged by the group's leader [27].

In addition to news reports, we identified nine articles published in academic journals, a single report on child marriage in North America published by a non-profit organization [42], and results from the 2016 Canadian Census available online from Statistics Canada (Table 1).

The information on early marriage contained in these articles is synthesized below. Our results are organized by estimates of the frequency of early marriage in Canada and the context of those marriages, followed by information on the motivations for early marriage and its consequences. We focus our discussion on the youngest age range available when sources presented relevant information for multiple age groups below the age of 25 (i.e. 15–19 and 20–24).

## How common is early marriage in Canada and what are the characteristics of persons who marry early?

Analyses of Canadian vital statistics and census data provide information on how common early marriage has been at various points in time. An examination of historic marriage patterns in Canada reported that in the late 1800s, 33% of women and 13% of men between 20 and 24 years of age had ever been married. Those proportions rose to 60% among women and 31% among men in 1961, and then fell again to 35% and 18% in 1991 [46]. Wadhera and Strachan (1992) used vital statistics data to examine trends in marriage rates across Canada between 1921 and 1990. They reported gender-specific rates among 15-19-year-olds between 1951 and 1990 and show that marriage among persons in this age range has become rarer over time.

Nationally, the rate of marriage among 15-19-year-old boys has declined steadily since 1971, from a high of 12.9 per 1000 in that year to 2.3 per 1000 in 1990. The rate of marriage among girls in this age range has been falling for a longer period of time, from a high of 71.5 per 1000 in 1956 to 10.6 per 1000 in 1990. The rate of marriage among teen girls was higher at all time points, and remained nearly 5 times higher in 1990. The observed decline in marriage rates among 15-19-year-olds over this period is consistent with the general trend toward delaying marriage in Canada, particularly among young women [48]. The same authors also published a detailed examination of marriage across Canada in 1991. Rates among 15-19-year-olds in 1991 do not appear to have changed substantially from those reported in 1990 [49].

More recently, the 2016 Canadian census recorded a total of 22,115 persons between 15 and 19 years of age who were married or in common law partnerships [43]. Using the total of 2,010,060 persons in that age range included in the census in that year as our denominator, we estimate that the prevalence of marriage, including common-law unions, among this age group was approximately 1.1% [43]. A breakdown of the number of marriages among 15-19-year-olds by province/territory and gender is also available, which permitted us to estimate the prevalence of early marriage, including common law unions, across the country. These results indicate that teens between 15 and 19 years of age residing in every province and territory were reported married or in common-law unions in 2016 (Table 2). Prevalence estimates among girls ranged from 1.0% to 2.7% percent in nearly all provinces and territories. The prevalence was lower among boys in every province and territory and ranged from 0.4% to 1.1%. Prevalence estimates from Nunavut suggest markedly different marriage patterns in the territory: 10.8% of girls and 6.3% of boys between 15 and 19 years of age were reportedly married or in common-law unions (Table 2). Similar results from earlier censuses may also be available, but novel analysis of trends over time is beyond the scope of this systematic review.

These academic sources provide estimates of how common marriage is among aggregate age groups within the country but information on precise age at marriage was not presented. We are unable to determine how many of the teens between 15 and 19 years of age married when they were 15, 16 or 17 years old. News sources document cases in which individual 16

**Table 2. Estimates of the prevalence of marriage, including common-law unions, among 15-19-year-olds by gender and province as recorded in the 2016 Canadian census [43].**

| Province/Territory | Number of 15-19-year-olds in residence | | Number of 15-19-year-olds married or in common-law union | | Prevalence of marriage, including common-law unions | |
|---|---|---|---|---|---|---|
| | *Females (a)* | *Males (b)* | *Females (c)* | *Males (d)* | *Females (c/a)* | *Males (d/b)* |
| Alberta | 114,680 | 121,775 | 2,385 | 860 | 2.1% | 0.7% |
| British Columbia | 125,180 | 132,015 | 1,915 | 765 | 1.5% | 0.6% |
| Manitoba | 39,115 | 41,660 | 805 | 320 | 2.1% | 0.8% |
| New Brunswick | 19,585 | 20,750 | 500 | 205 | 2.6% | 1.0% |
| Newfoundland and Labrador | 13,230 | 13,930 | 240 | 90 | 1.8% | 0.6% |
| Northwest Territories | 1,275 | 1,375 | 35 | 15 | 2.7% | 1.1% |
| Nova Scotia | 24,805 | 26,025 | 470 | 190 | 1.9% | 0.7% |
| Nunavut | 1,530 | 1,595 | 165 | 100 | 10.8% | 6.3% |
| Ontario | 393,140 | 414,425 | 3,965 | 1,660 | 1.0% | 0.4% |
| Prince Edward Island | 4,160 | 4,355 | 70 | 35 | 1.7% | 0.8% |
| Quebec | 209,455 | 217,615 | 4,405 | 1,655 | 2.1% | 0.8% |
| Saskatchewan | 32,450 | 34,040 | 870 | 350 | 2.7% | 1.0% |
| Yukon | 915 | 975 | 20 | 10 | 2.2% | 1.0% |

Source: Statistics Canada, 2016 Census of Population, Statistics Canada Catalogue no. 98-400-X2016028

and 17-year old girls were married and the Calgary Herald reported that twenty-nine children below the age of 15 were married in Canada in 1982: 26 girls and 3 boys [40]. However, reports from small religious communities may not reflect broader trends in age at marriage and the Calgary Herald report reflects marriages that occurred over 35 years ago. Marriage before the age of 16 has been prohibited in Canada since 2015 [22].

Other sources provide limited insight into the context of early marriages in Canada. Gentleman and Park examined age differences between spouses captured in the 1991 census. The age difference between spouses when the wife was between 15 and 20 years old was larger than in any other age bracket: the median age difference was 3.9 years, nearly 2 years greater than the median difference in any other age range, and in 3.6% of these couples the husband was at least 15 years older than his wife, a larger percentage than in any other age category [47]. This indicates that the youngest brides tend to marry older men relative to those who marry later in life. Divorce rates were also higher among couples with a larger age difference. This is consistent with another study that showed some teen marriages in Canada were short-lived. Lapierre (1991) found that approximately 750 of every 100,000 married boys and 1000 of every 100,000 married girls between 15 and 19 years of age were granted divorces in 1990 [50].

## Motivations for early marriage

Our results tell us little about the motivations for early marriage in Canada. News reports of early marriages among diverse religious communities suggest that religious beliefs may encourage early marriage, though the precise nature of these beliefs were not discussed in detail [26, 27, 28]. Statements from two women who married at 16 years of age in the Fundamentalist Latter Day Saints community in Bountiful, British Columbia, indicated that their educational goals played a role in the timing of their marriages. According to those statements, leaders of the community would only permit women to further their education if they were married [26, 28]. However, the motivations described in these individual cases that occurred among small and distinct communities may not represent those among the broader population.

Sen and Ariizumi showed that increases in the minimum wage across Canada between 1990 and 2005 were associated with increases in marriage rates and births among 15-19-year-olds, but the mechanisms through which changes to the minimum wage may influence these life events are unclear [44]. The authors hypothesize that a rising minimum wage may have reduced the opportunity cost of having a child, thereby increasing teen pregnancy rates that may have led to marriages, or that the potential for greater future earnings may have encouraged teen marriages. The provincial-level data used to generate these results severely limits the extent to which they can be used to understand individual-level motivations for marriage. Moreover, this relationship may differ for marriages before the age of 18, which are less common than those among 18 and 19-year-olds and unique in that they require parental consent.

## Consequences of early marriage

Information on the consequences of early marriages in Canada is also limited. Grindstaff (1988) reported that women who married before the age of 20 and were 30 years of age at the time of the 1981 Canadian census had lower educational attainment, on average, when compared to their peers who married at later ages. More than half of women who married before 20 years of age had not completed secondary school. They were also more likely to be employed in the service-sector and reported lower income. Though the report by Grindstaff indicates strong associations between marrying as a teen and economic outcomes at the age of 30, no attempt was made to control for other characteristics that may influence age at marriage and these outcomes, which severely limits the strength of the conclusions that can be drawn.

We cannot conclude that these results indicate that early marriage caused unintended early ends to education for these women; it is also possible that early marriage was perceived as an alternative pathway to adulthood for some who did not intend or desire to further their schooling. Case reports also indicate that limited educational opportunities following early marriage and the birth of children may lead to financial dependence upon a spouse, though women who participated in interviews with Morin (2017) expressed different opinions regarding whether this dependence was of concern to them.

Statements from two women who married at very young ages indicate that they experienced abuse within their marriages [33, 38]. However, the exclusive focus of news reports on child marriage among religious minority communities may severely restrict the generalizability of the experiences highlighted. A single study based on a small cohort of persons who graduated from high school in Edmonton, Alberta in 1985 found that women who married before the age of 23 reported experiencing more depressive symptoms in their early 40s when compared to those who married at later ages [41]. However, fewer than half of the original cohort was included in the study, leading to concerns regarding selection bias, and the covariates included in the analyses were unlikely to limit confounding of the relationship between early marriage and mental illness.

## Conclusions

Published results from historical studies and a recent census provide information on the frequency of marriage among teens between 15 and 19 years of age in Canada. The prevalence of marriage among this age group varies across the country but the reasons for that variation are unknown and have received little attention. In particular, the relatively high prevalence of marriage among teens in Nunavut warrants further investigation. Family formation patterns may differ in the territory, where more than 85% of the population identified as Indigenous in the 2016 census, relative to other parts of the country [52]. The severe housing shortage may also be a contributing factor. In 2016, more than half of Nunavummiut were living in social housing and overcrowding within social housing remains an urgent concern [53]. Young couples may marry or enter common law unions at early ages if marriage speeds access to adequate housing. Future research led by communities in Nunavut may shed light on why early marriage is more common there and whether early marriage is perceived as problematic among Nunavummiut.

Aggregation over a five-year age range in the census and in other academic studies prevented us from using this information to assess the frequency of marriage before the age of 18 in Canada. We found no published studies that included such estimates. Our results indicate that the marriage of persons under the age of 18 has received almost no direct attention outside of media reports on individual cases among religious minority communities. Academic reports on marital trends have not drawn particular attention to the earliest age categories nor do they suggest that there is reason for concern regarding those marriages, even when they document that the context of marriages among 15-19-year-olds differs from marriages among older persons [47]. This is curious given the country's explicit attention to and concern regarding marriages before the age of 18 internationally and suggests that the same perception of early marriage as an indicator of gender equality and adolescent health is not being applied domestically.

Although marriage before the age of 18 has not been studied systematically in Canada, available information on marriages to 15-19-year-olds in the country indicate that these marriages follow patterns similar to those observed among children in other parts of the world. Strong evidence of the gendered nature of early marriage in Canada is consistent across sources and time periods: all of the nationwide estimates we identified, from the late 1800s through 2016, show that girls are more likely to be married at young ages than boys. We found no case reports of boys married before the age of 18 and the limited evidence of the potential

harms of early marriage is also largely focused on women and girls [33, 51]. Evidence from the 1990s also shows that married 15-19-year-old girls in Canada were wed to substantially older spouses relative to women who married at older ages [47]. This is consistent with evidence from low and middle-income countries and gives rise to concerns regarding the power dynamics within highly age-discrepant marriages [8].

The health-related consequences of early marriage have received very little attention in Canada and the existing evidence is too weak to draw conclusions. This is in stark contrast to literature on child marriage in other regions of the world, which often portrays the negative impact on girls' health as one of the foremost reasons for concern regarding early marriage [6, 7, 10, 12, 54]. The reasons for this lack of attention are unclear but may stem from the perception that child marriage is something that only happens elsewhere or among small and culturally distinct populations within Canada. More than half of the articles we identified through this systematic review were news reports of child marriages that occurred among religious minority communities. The narrow focus of this media coverage may suggest to readers that the marriage of young girls in Canada occurs primarily among groups whose beliefs and practices may be far removed from the experience of most Canadians. However, estimates from other sources including the 2016 census indicate that early marriages occur nationwide and at larger scale. Focusing on religious minority communities may neglect the larger picture of child marriage in Canada.

Estimates of the incidence and prevalence of marriage among children under the age of 18 in Canada are needed so that the country can assess its own progress toward the United Nations Sustainable Development Goals. Aggregate marriage statistics among 15-19-year-olds fail to make a distinction between teen minors, who require parental permission to marry and whose autonomy is restricted in many other realms, and teens who are over the age of majority and considered legal adults. Disaggregation based on this importation social distinction would have been more informative in the case of marriage statistics and likely in many others. We take this opportunity to encourage thoughtful consideration of the use of aggregate statistics and to call for disaggregation when possible.

Transparent reporting on child marriage practices in Canada would acknowledge that it is a global issue and not restricted to certain regions of the world. Additional research on the motivations for and consequences of child marriage in Canada would also facilitate needed discussion regarding the discrepancy between domestic laws that continue to permit children under the age of 18 to wed and Canada's international commitments to end the practice. Such a discourse may shed light on the remaining barriers to eliminating child marriage and strengthen the country's efforts to improve gender equality and support the rights of women and girls both domestically and internationally.'

## Supporting information

**S1 Checklist. Supporting information: Preferred Reporting Items for Systematic Reviews and Meta-Analyses (PRISMA) checklist.**
(DOC)

**S1 File. Supporting information: Details of database search terms.**
(DOCX)

## Author Contributions

**Conceptualization:** Alissa Koski.

**Data curation:** Michele Zaman, Alissa Koski.

**Formal analysis:** Michele Zaman, Alissa Koski.

**Funding acquisition:** Alissa Koski.

**Methodology:** Michele Zaman, Alissa Koski.

**Project administration:** Alissa Koski.

**Resources:** Alissa Koski.

**Supervision:** Alissa Koski.

**Writing – original draft:** Michele Zaman, Alissa Koski.

**Writing – review & editing:** Michele Zaman, Alissa Koski.

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
