## [Decision Letter · Decision Letter 0]

13 Jan 2020

PONE-D-19-31231

Child Marriage in Canada: A Systematic Review

PLOS ONE

Dear Ms Zaman,

Thank you for submitting your manuscript to PLOS ONE. After careful consideration, we feel that it has merit but does not fully meet PLOS ONE’s publication criteria as it currently stands. Therefore, we invite you to submit a revised version of the manuscript that addresses the points raised during the review process.

ACADEMIC EDITOR:  This is a great paper and only requires a few small edits before it can be accepted. My feedback is provided below. Thank you for your submission. 

We would appreciate receiving your revised manuscript by Feb 27 2020 11:59PM. To enhance the reproducibility of your results, we recommend that if applicable you deposit your laboratory protocols in protocols.io, where a protocol can be assigned its own identifier (DOI) such that it can be cited independently in the future. For instructions see: http://journals.plos.org/plosone/s/submission-guidelines#loc-laboratory-protocols

We look forward to receiving your revised manuscript.

Kind regards,

David W Lawson

Academic Editor

PLOS ONE

Journal Requirements:

2. Please ensure that you include a title page within your main document. You should list all authors and all affiliations as per our author instructions and clearly indicate the corresponding author.

Additional Editor Comments:

This is a great paper! Clear in scope, well-written and the conclusions justified. It makes a very useful contribution to the literature on early marriage! Thank you for the opportunity to read and review your work.

The reviewers have a few minor concerns, which I agree with. I would like to see these points addressed before officially accepting the paper. This is all small potatoes stuff and should not take long to do.

I would add to this a general concern that you do need to be more explicit throughout - perhaps especially in your introduction - that demonstrated associations between early marriage and negative outcomes for girls/women may not always be because early marriage causes those negative outcomes. To make a parallel - the teen pregnancy literature has over time walked back from some of the early strong conclusions that teen marriage is an ultimate driver of poor wellbeing for young women, and recognized that teen pregnancy often is better understood as a product of low opportunity / lack of options. Similar shifts are afoot with the child marriage literature, especially in settings where it mainly takes place in late adolescence, and your framing would benefit from reflecting this - rather than reiterating stylized truths about the universal harms of child marriage.

Some relevant manuscripts which make these points:

- On teen pregnancy:

Furstenberg, F. (2016). Reconsidering teenage pregnancy and parenthood. Societies, 6(4), 33.

- On child marriage: Stark, L. (2018). Poverty, consent, and choice in early marriage: ethnographic perspectives from urban Tanzania. Marriage & Family Review, 54(6), 565-581.

- Schaffnit, S. B., Hassan, A., Urassa, M., & Lawson, D. W. (2019). Parent–offspring conflict unlikely to explain ‘child marriage’in northwestern Tanzania. Nature human behaviour, 3(4), 346.

Reviewers' comments:

Reviewer's Responses to Questions

**Comments to the Author**

1. Is the manuscript technically sound, and do the data support the conclusions?

Reviewer #1: Yes

Reviewer #2: Yes

2. Has the statistical analysis been performed appropriately and rigorously? 

Reviewer #1: N/A

Reviewer #2: N/A

3. Have the authors made all data underlying the findings in their manuscript fully available?

Reviewer #1: Yes

Reviewer #2: Yes

4. Is the manuscript presented in an intelligible fashion and written in standard English?

Reviewer #1: Yes

Reviewer #2: Yes

5. Review Comments to the Author

Reviewer #1: This manuscript was clear, thoughtfully written and a pleasure to read. The research design was solid. I have a few comments aimed at improvement, however.

Line 35. The word 'controlled' here sounds a bit patronizing. Controlled by whom? This could be said in a different way.

Lines 46-47, it should be made clear that this is not necessarily a causal relationship, unless you state otherwise.

Lines 49-53. Same here, which causes which? A growing body of research is suggesting that a lack of affordable and decent quality education in fact leads to early marriage, or that they are co-causal.

Lines 177-178: This information about Canadian law should come earlier in the manuscript, at the beginning.

Line 185: This is one of your conclusions, please make sure it is clearly mentioned there.

Lines 206-209: This is interesting, and perhaps should be highlighted a bit more clearly in your text.

Line 209: I am not sure here what you mean by the word 'ecological'.

The 'Conclusion' section is for the most part very good and makes a useful contribution to the field, but I would ask the authors to expand on two points within it:

First, the high prevalence of early marriage among teens in Nunavut is quite eye-catching in the data section where it is reported. Do the authors not have any 'best guesses' about the reasons behind this? Surely some informed speculation could be ventured here to guide future researchers.

Second, the authors usefully note that aggregation within their data presents difficulties in drawing further conclusions. This could be the springboard for a recommendation for future research: that it should always be disaggregated by age (as well as socio-economic status and so forth). The problem of aggregation affects a broad range of disciplines within the social sciences, so it is always a useful reminder to other researchers to avoid it where possible.

Reviewer #2: This is a nicely written paper and clearly makes the case that more research is needed on early marriage in Canada. My one minor comment has to do with the introduction. I would like to see the causal link between early marriage and negative health related outcomes (paragraph 2) questioned or at least a short note suggesting that correlations could be due to selection effects rather than harmful consequences of early marriage per se.

I enjoyed the discussion. It was fair and clearly identified gaps in knowledge.

6. PLOS authors have the option to publish the peer review history of their article (what does this mean?). If published, this will include your full peer review and any attached files.

Reviewer #1: Yes: Laura Stark

Reviewer #2: No

---

## [Author Response · Author response to Decision Letter 0]

6 Feb 2020

We have responded to the editor and peer-reviewer comments in a letter which is included with this submission. Thank you!

---

## [Editor Report · Decision Letter 1]

12 Feb 2020

Child Marriage in Canada: A Systematic Review

PONE-D-19-31231R1

Dear Dr. Koski,

We are pleased to inform you that your manuscript has been judged scientifically suitable for publication and will be formally accepted for publication once it complies with all outstanding technical requirements.

With kind regards,

David W Lawson

Academic Editor

PLOS ONE

Additional Editor Comments (optional):

Thank you for your thoughtful revisions. This is great paper and an important contribution to the literature on early / child marriage.

Thank you for the opportunity to review your work.

---

## [Editor Report · Acceptance letter]

20 Feb 2020

PONE-D-19-31231R1 

Child Marriage in Canada: A Systematic Review 

Dear Dr. Koski:

I am pleased to inform you that your manuscript has been deemed suitable for publication in PLOS ONE. Congratulations! Your manuscript is now with our production department. 

With kind regards,

on behalf of

Dr. David W Lawson 

Academic Editor

PLOS ONE